# Bias in Zeroth-Order Normal Estimation for Decision-Based Attacks

**Feiyang Wang** [1 2]   **Hangwei Qian** [3]   **Xingquan Zuo** [1 2]   **Gang Chen** [4]   **Ivor Tsang** [3 5]

## Abstract

Decision-based image attacks commonly rely on zeroth-order (ZO) Monte Carlo probing to estimate decision-boundary normals and iteratively refine adversarial perturbations to minimize the $\ell_2$ norm. We theoretically analyze and empirically demonstrate an *intrinsic inefficiency* arising from *heterogeneous input sensitivity*, where only a small subset of coordinates strongly affects the target model's predictions. Empirically, with one-bit feedback and a limited query budget, updates on low-sensitivity coordinates are overwhelmed by initialization and sampling noise, preventing their perturbations from exhibiting consistent improvement. By modeling ZO refinement as a stochastic dynamical system, we formally characterize its asymptotic behavior: the optimization enters a stationary regime, where the perturbation aligns (in expectation) with the normal and its coordinate-wise magnitudes encode a local sensitivity ranking. However, *this stationarity does not generally yield $\ell_2$-optimal perturbations under nonlinear boundaries*. Building on this observation, we propose a novel and effective algorithm, *Sensitivity-Aware Rescaling* (SAR), that leverages this sensitivity signal to infer an importance map from the current best perturbation, then progressively suppresses low-importance regions through a coarse-to-fine schedule to reduce the $\ell_2$ norm. Extensive experiments show that SAR achieves consistent improvements in perturbation norm, attack success rate, and visual imperceptibility. The code is available at `https://github.com/Flyingssheep/SAR`.

---

[1]School of Computer Science, Beijing University of Posts and Telecommunications (BUPT), China [2]Key Laboratory of Trustworthy Distributed Computing and Services, China [3]CFAR and IHPC, Agency for Science, Technology and Research (A*STAR), Singapore [4]Victoria University of Wellington, New Zealand [5]Nanyang Technological University (NTU), Singapore. Correspondence to: Xingquan Zuo <zuoxq@bupt.edu.cn>, Hangwei Qian <Qian_Hangwei@a-star.edu.sg>.

*Proceedings of the $43^{rd}$ International Conference on Machine Learning*, Seoul, South Korea. PMLR 306, 2026. Copyright 2026 by the author(s).

## 1. Introduction

Deep neural networks now underpin high-impact systems in vision, speech, and language (Wang et al., 2024). However, their predictions can often be manipulated by adversarial examples: inputs with small, often imperceptible perturbations that induce misclassifications (Carlini & Wagner, 2017; Cheng et al., 2019; Sun et al., 2024). This vulnerability raises practical security and safety concerns in settings such as biometric authentication, medical imaging, and autonomous perception (Chen et al., 2017; Fan et al., 2024).

**Decision-based (hard-label) black-box attacks** represent one of the most restrictive yet realistic threat models (Brendel et al., 2018). In this setting, an attacker interacts with the target model through *one-bit feedback* where each query reveals only the final predicted label, with no access to confidence scores, gradients, or internal model parameters. This hard-label constraint reflects the operating conditions of many real-world black-box APIs (e.g., commercial image-recognition services), making query-efficient decision-based attacks a central challenge in adversarial machine learning (Cheng et al., 2020; Li et al., 2021; Yang et al., 2024).

Most state-of-the-art decision-based attacks follow a common paradigm: **zeroth-order (ZO) decision-boundary normal estimation coupled with iterative refinement** (Chen et al., 2020a; Rahmati et al., 2020; Reza et al., 2023). *ZO decision-boundary normal estimation* aims to estimate the normal vector of the decision boundary using only one-bit hard-label feedback. In practice, the decision-boundary normal is typically estimated via Monte Carlo (MC) directional probing (see Section 4.1); e.g., (Chen et al., 2020a; Rahmati et al., 2020; Reza et al., 2023). Methods such as HSJA and CGBA instantiate this MC-based routine and iteratively refine a candidate adversarial example under a tight query budget (Brendel et al., 2018; Chen & Gu, 2020; Bai et al., 2023). Despite their empirical success, the mechanisms governing both the efficiency and the failure modes of this paradigm have not been systematically analyzed.

Specifically, ZO normal-based attacks implicitly assume a *dimensionally homogeneous* input space, treating all coordinates as equally informative during refinement (Chen et al., 2020a). In many real-world images, however, model sensitivity is highly heterogeneous: only a small subset of pixels or regions strongly influences the decision, while large back-

ground areas contribute negligibly (Selvaraju et al., 2017). In this work, we show that such heterogeneity creates an **unavoidable, structural bias** in ZO normal-based attacks under one-bit feedback: the hard-label constraint effectively limits the usable dimensionality under a finite query budget. Hence, low-sensitivity coordinates receive vanishing expected updates, leading to persistently suboptimal $\ell_2$ distortions. Crucially, this limitation arises at the level of expectation and is intrinsic to hard-label feedback, rather than a variance effect that could be alleviated by increasing Monte Carlo samples or tuning hyperparameters (Mohamed et al., 2020). As a result, perturbation mass can persist on non-informative dimensions, inflating the overall $\ell_2$ distortion and degrading perceptual quality.

Through in-depth analysis of this critical bias, we derive dimension-wise characterizations of the mean, variance, and signal-to-noise ratio of the perturbation updates, then uncover a practically important **sensitivity-encoding property**: **the stationary perturbation inherently encodes sensitivity**, i.e., the final perturbation produced by ZO normal-based attacks exhibits a clear coordinate-wise magnitude pattern that matches the underlying sensitivity ranking. We formalize this observation by modeling ZO normal-based updates as a stochastic dynamical system. To our knowledge, these results reveal for the first time in literature how input sensitivity governs systematic optimization drift.

Leveraging the sensitivity-encoding property, we propose a novel **Sensitivity-Aware Rescaling (SAR)** algorithm, a lightweight plug-in refinement method that can be seamlessly integrated into existing ZO normal-based decision attacks without altering their core pipelines. SAR constructs an importance map from the current best boundary perturbation, and applies conservative suppression to inferred non-sensitive regions under a coarse-to-fine schedule. This design directly targets stagnation on low-sensitivity dimensions in the hard-label setting. We evaluate SAR on standard benchmarks and across multiple decision-based attack methods. The proposed SAR algorithm consistently reduces perturbation norms, improves visual imperceptibility, and noticeably increases attack success rates. Notably, heterogeneous input sensitivity emerges as a fundamental driver of the optimization dynamics in decision-based attacks under one-bit feedback. Recognizing this effect therefore shifts the analysis from descriptive observations to a core methodological issue, revealing an inherent structural limitation of existing approaches and establishing a principled foundation for designing substantially more efficient attacks.

## 2. Related Work

**Decision-based attacks.** Existing decision-based attacks can be largely divided into two categories: *randomized search* and *ZO normal estimation* (Li et al., 2021). Random-ized search attacks explore the input space stochastically and accept candidate perturbations only when they reduce the decision boundary distance, e.g., Boundary Attack and its variants (Brendel et al., 2018; Brunner et al., 2019; Li et al., 2021), SurFree (Maho et al., 2021), Triangle Attack (Wang et al., 2022), and progressive search strategies such as RayS (Chen & Gu, 2020) and ADBA (Wang et al., 2025b).

In contrast, *ZO normal-based attacks* can often achieve state-of-the-art performance under the $\ell_2$ norm by explic-itly estimating the local boundary normal vector and up-dating perturbations accordingly (Liu et al., 2019; Chen et al., 2020a; Rahmati et al., 2020; Li et al., 2020; Ma et al., 2021; Wan et al., 2024; Reza et al., 2023). A representa-tive pipeline typically follows three steps. (i) Project an initial adversarial example onto the decision boundary via binary search. (ii) Estimate the local boundary normal by querying perturbed samples around the boundary point. (iii) Update the adversarial perturbation by moving along the estimated normal direction and re-project it onto the deci-sion boundary, iteratively shrinking the $\ell_2$ distance (Chen et al., 2020a; Ma et al., 2021; Wan et al., 2024). Recent works such as HSJA (Chen et al., 2020a), CGBA (Reza et al., 2023), GOBA (Yang et al., 2024), and TtBA (Wang et al., 2025a) enhance boundary navigation through struc-tured low-dimensional search (e.g., semicircular search) or improved theoretical analysis, while still treating accurate normal estimation as the central primitive.

**Bias, Theoretical Gaps, and Region Sensitivity.** Exist-ing theories for decision-based attacks have mainly focused on explaining how local decision-boundary regularity (e.g., smoothness or low curvature) enables efficient zeroth-order normal estimation, and why normal-based updates com-bined with projection can effectively reduce the $\ell_2$ distor-tion (Liu et al., 2019; Rahmati et al., 2020; Chen et al., 2020a). GOBA (Yang et al., 2024) provides a rigorous vari-ance analysis of the MC normal estimator, revealing how its stochastic error affects the optimization trajectory. Guessing Smart (Brunner et al., 2019) and SDR (Sun et al., 2022) in-troduce heuristic sampling biases to steer exploration away from ineffective directions; however, they do not provide a principled mechanism for identifying and filtering unin-formative dimensions. Critically, existing analyses assume isotropic, uniformly informative sampling, overlooking a key limitation: with heterogeneous pixel-wise sensitivity, normal estimation produces inherently uneven and often ineffective dimension-wise refinement signals. Meanwhile, region-sensitive or sparse perturbation methods based on saliency, surrogate guidance, or coordinate/patch search (Dong et al., 2020; Chen et al., 2020b; Lin et al., 2023; Shi et al., 2022; Tao et al., 2023) are largely heuristic and of-ten rely on additional assumptions or extra queries. They hence fail to establish the fundamental necessity of sensitive-

region refinement under strict hard-label constraints.

In this paper, we present the first theoretical result showing that, under heterogeneous input sensitivity, low-sensitivity dimensions have zero expected drift in ZO normal estimation. Therefore, non-informative coordinates cannot effectively drive the optimization process, revealing an inherent *expectation-level bias*.

## 3. Problem Definition

**Decision-based attack.** Let $x \in [0,1]^D$ denote a source image, where $D = C \times W \times H$ is the dimensionality of the flattened image, and let $y(x) \in \{1, \ldots, K\}$ be its ground-truth label. Consider a $K$-class image classifier that outputs confidence scores $f_k(x) \in \mathbb{R}$ for each class $k \in \{1, \ldots, K\}$, and predicts the label $f(x) = \arg\max_k f_k(x)$. We assume that the source image $x$ is correctly classified, i.e., $f(x) = y(x)$. The goal of a decision-based attacker is to find an adversarial example $\tilde{x} \in [0,1]^D$ by solving

$$\min_{\tilde{x} \in [0,1]^D} \|\tilde{x} - x\|_2 \quad \text{s.t.} \quad I(\tilde{x}) = 1, \tag{1}$$

where $I(\cdot)$ is a hard-label indicator function defined as

$$I(\tilde{x}) = \begin{cases} 1, & \text{if } f(\tilde{x}) \neq y(x), \\ -1, & \text{otherwise.} \end{cases} \tag{2}$$

The hard-label indicator can be equivalently induced by a score-based decision function (introduced for analysis)

$$\phi(\tilde{x}) = \max_{k \neq y(x)} f_k(\tilde{x}) - f_{y(x)}(\tilde{x}), \tag{3}$$

by utilizing $I(\tilde{x}) = \text{sign}(\phi(\tilde{x}))$.

**Perturbation direction.** Let $d \in \mathbb{R}^D$ denote a perturbation direction. For any radius (i.e., perturbation strength) $r \geq 0$, we define the perturbed input along direction $d$ as

$$\tilde{x}(r; d) = \text{clamp}(x + r \cdot d/\|d\|_2), \tag{4}$$

where $\text{clamp}(\cdot)$ truncates each pixel into $[0,1]$.

We define the *boundary distance* along direction $d$ as the minimum radius required to reach the adversarial region:

$$g(d) = \inf \{r > 0 \ : \ I(\tilde{x}(r; d)) = 1\}. \tag{5}$$

Accordingly, the corresponding boundary point along direction $d$ is defined as

$$\tilde{x}^*(d) = \tilde{x}(g(d); d) = \text{clamp}(x + g(d) \cdot d/\|d\|_2). \tag{6}$$

Under the assumption that the adversarial region is closed along the search ray, $\tilde{x}^*(d)$ lies on the boundary, satisfying

$$I(\tilde{x}^*(d)) = 1 \quad \text{and} \quad I(\tilde{x}^*(d) - \epsilon \cdot d) = -1, \tag{7}$$

for any sufficiently small $\epsilon > 0$. With this definition, Eq. (1) can be reformulated as

$$\min_{d \neq 0} g(d) \quad \text{s.t.} \quad I(\tilde{x}(g(d); d)) = 1. \tag{8}$$

## 4. Analysis of Zeroth-Order Decision Attacks

This section conducts an in-depth theoretical analysis of a broad class of decision-based attacks driven by ZO estimations of decision-boundary normals. We first formalize a pixel-wise sensitivity profile and then model the standard MC normal-driven perturbation optimization pipeline as a stochastic dynamical system (Section 4.1). Under standard local assumptions, Lemma 4.1 below characterizes the coordinate-wise mean and variance of the MC normal estimator, enabling a direct signal-to-noise analysis. Building on this, Proposition 4.2 further reveals an *effective dimensionality constrained by the query budget*: with $N$ hard-label queries per iteration, only $O(N)$ coordinates exhibit non-negligible expected update, while the remaining coordinates are dominated by stochastic fluctuations. Finally, theorem 4.3 specifically shows that any directionally stationary perturbation aligns with the boundary normal in expectation and therefore *encodes* the local sensitivity ranking via its coordinate-wise magnitudes. These insights motivate us to develop a novel sensitivity-aware rescaling algorithm, **SAR**, which can be integrated into existing attack methods to further reduce the $\ell_2$ perturbation norm. All technical proofs are deferred to appendix A.

### 4.1. Preliminaries

**Pixel-wise perturbation sensitivity.** To evaluate the effectiveness of refining an existing adversarial example across different pixel dimensions, we consider the decision function $\phi(\tilde{x})$ in Eq. (3) and define its local gradient at a perturbed point $\tilde{x} \in \mathbb{R}^D$ as

$$\nabla\phi(\tilde{x}) := \left[\frac{\partial\phi(\tilde{x})}{\partial\tilde{x}_1}, \ldots, \frac{\partial\phi(\tilde{x})}{\partial\tilde{x}_D}\right]^\top, \tag{9}$$

with the $i$-th component $\nabla_i\phi(\tilde{x}) := \frac{\partial\phi(\tilde{x})}{\partial\tilde{x}_i}$ ($i$ is the pixel index). We define the pixel-wise perturbation sensitivity as

$$s_i(\tilde{x}) := |\nabla_i\phi(\tilde{x})|, \quad i = 1, \ldots, D. \tag{10}$$

Substituting $\phi(\tilde{x}) = \max_{k \neq y(x)} f_k(\tilde{x}) - f_{y(x)}(\tilde{x})$ and assuming a unique maximizer $k^* \in \arg\max_{k \neq y(x)} f_k(\tilde{x})$ (holding almost everywhere), we have

$$s_i(\tilde{x}) = |\nabla_i f_{k^*}(\tilde{x}) - \nabla_i f_{y(x)}(\tilde{x})|. \tag{11}$$

Intuitively, if the sensitivity items $\{s_i(\tilde{x})\}$ are directly accessible, one could explicitly emphasize high-sensitivity dimensions, enabling more effective refinement of the current perturbation.

**ZO normal estimation.** In the decision-based black-box setting, however, neither the score functions $\{f_k(\cdot)\}_{k=1}^K$ nor the perturbation sensitivity $\{s_i(\cdot)\}$ is directly accessible. A common approach is to estimate the local decision boundary

normal using Monte Carlo (MC) directional probing with hard-label queries (Chen et al., 2020a; Reza et al., 2023; Yang et al., 2024; Wang et al., 2025a): given a boundary point $\tilde{x}^*(d)$ defined in Eq. (6), sample $N$ random unit directions $\{u_i\}_{i=1}^N$ i.i.d. from the uniform distribution over the unit sphere $\mathbb{S}^{D-1}$ in $\mathbb{R}^D$, i.e., $u_i \sim \mathrm{Unif}(\mathbb{S}^{D-1})$. With a small probing step size $\delta > 0$, the normal is estimated by

$$\hat{n}(\tilde{x}^*(d)) = \frac{1}{N} \sum_{k=1}^N I(\tilde{x}^*(d) + \delta u_k) \cdot u_k. \qquad (12)$$

Eq. (12) forms a sign-weighted average of random unit directions, where the sign is determined solely by hard-label feedback. The resulting vector $\hat{n}$ estimates the local decision-boundary normal and provides the optimization direction in decision-based attacks.

**Normal-driven perturbation optimization.** Using the estimated normal $\hat{n}$, MC normal-based attacks iteratively refine the perturbation direction $d$ to reduce the boundary distance $g(d)$. Each iteration consists of two steps: updating the direction using $\hat{n}$, followed by projecting the result back onto the decision boundary (Wang et al., 2025a). Starting from an initial direction $d^{(0)}$, the update at iteration $t$ is

$$d^{(t+1)} = d^{(t)} + \alpha_t \cdot \hat{n}(\tilde{x}^*(d^{(t)})), \qquad (13)$$

$$\tilde{x}^{(t+1)} = \tilde{x}^*(d^{(t+1)}) = \tilde{x}(g(d^{(t+1)}); d^{(t+1)}), \qquad (14)$$

where $\alpha_t > 0$ is a step size decided by attackers. Each iteration therefore consists of a direction update followed by a boundary projection, which together progressively reduce the decision boundary distance $g(d)$.

**Limitations.** Based on the sensitivity profile defined in Eq. (10), we theoretically show below that when $\{s_i(\tilde{x})\}_{i=1}^D$ is heterogeneous, the isotropic MC normal estimation in Eq. (12) can become systematically biased, causing low-sensitivity dimensions to be under-represented in the estimated normal vector $\hat{n}$.

## 4.2. Main Theoretical Results

**Assumptions.** We summarize all assumptions used in our theoretical analysis, these assumptions are widely adopted in decision-based black-box analysis:

1. (**Local linear sign model**) For a sufficiently small probing radius $\delta > 0$, $I(\tilde{x}^* + \delta u) \approx \mathrm{sign}(\nabla\phi(\tilde{x}^*)^\top u)$.

2. (**Uniform random directions**) The Monte Carlo directions are i.i.d. $u \sim \mathrm{Unif}(\mathbb{S}^{D-1})$.

3. (**Large-$D$ approximation**) $D$ is assumed to be large such that $c_D \approx \sqrt{2/(\pi D)}$.

4. (**Robbins–Monro step sizes**) In Eq. (13), the step sizes satisfy $\alpha_t > 0$, $\sum_t \alpha_t = \infty$, and $\sum_t \alpha_t^2 < \infty$.

Assumption 1 captures the common local linearization of the decision boundary under small probing radius $\delta$, which is routinely used when analyzing hard-label queries (Shi et al., 2022). Assumption 2 is the default isotropic sampling scheme in MC normal estimation (Chen et al., 2020a; Reza et al., 2023; Wang et al., 2025a). Assumption 3 is used only to reliably approximate $c_D$ asymptotically, which is a standard practice for high-dimensional inputs such as images (Jameson, 2013). Finally, Assumption 4 is a classical step-size condition in stochastic approximation, ensuring stable updates in expectation (Borkar & Meyn, 2000).

**Lemma 4.1** (Mean and variance of MC normal components under local linearity)**.** *Consider the MC estimator in Eq. (12). Then for each coordinate/pixel $i = 1, \ldots, D$,*

$$\mathbb{E}[\hat{n}_i] \approx c_D \frac{\nabla_i \phi(\tilde{x}^*)}{\|\nabla\phi(\tilde{x}^*)\|_2}, \qquad (15)$$

$$\mathrm{Var}(\hat{n}_i) \approx \frac{1}{N}\left(\frac{1}{D} - c_D^2 \frac{\nabla_i\phi(\tilde{x}^*)^2}{\|\nabla\phi(\tilde{x}^*)\|_2^2}\right), \qquad (16)$$

*where the dimension-dependent constant $c_D$ is*

$$c_D := \mathbb{E}_u\left[\left|\left(\frac{\nabla\phi(\tilde{x}^*)}{\|\nabla\phi(\tilde{x}^*)\|_2}\right)^\top u\right|\right] = \frac{\Gamma(\frac{D}{2})}{\sqrt{\pi}\Gamma(\frac{D+1}{2})}, \qquad (17)$$

$\Gamma(\cdot)$ *denotes the Gamma function. Moreover, for large $D$,*

$$c_D \approx \sqrt{\frac{2}{\pi D}}. \qquad (18)$$

*Consequently, for any $i, j$ with $\mathbb{E}[\hat{n}_j] \neq 0$,*

$$\frac{|\mathbb{E}[\hat{n}_i]|}{|\mathbb{E}[\hat{n}_j]|} \approx \frac{|\nabla_i\phi(\tilde{x}^*)|}{|\nabla_j\phi(\tilde{x}^*)|} = \frac{s_i(\tilde{x}^*)}{s_j(\tilde{x}^*)}. \qquad (19)$$

**Summary of Lemma 4.1.** This lemma is proved in Appendix A.1. This lemma characterizes the mean and variance properties of the MC estimator for the decision boundary normal vector $\hat{n}$. Specifically, each component's expectation $\mathbb{E}[\hat{n}_i]$ aligns with the normalized gradient direction $\nabla_i\phi/\|\nabla\phi\|_2$, whereas its variance depends jointly on the total dimension $D$ and the number of MC samples $N$.

Crucially, the lemma shows that the relative magnitudes of the expected estimator components directly correspond to the true pixel-wise sensitivities $s_i = |\nabla_i\phi|$ (Eq. (19)). This provides the theoretical foundation for inferring the relative importance of different pixel dimensions through hard-label queries alone, without relying on gradient information.

Building on Lemma 4.1, we next translate these moment characterizations into an optimization-relevant quantity: the per-coordinate signal-to-noise ratio (SNR), which shows that low-sensitivity coordinates are dominated by noise.

**Proposition 4.2** (Initialization-dominated coordinates and $O(N)$ effective refinement). *At the $t$-th iteration and at the current boundary point $\tilde{x}^*$, consider the ZO normal-estimation refinement update in Eq. (13) where $\hat{n}$ is the MC estimator of the normal vector constructed from $N$ random directions. Then the refinement of dimension $i$ is governed by the per-iteration signal-to-noise ratio*

$$\text{SNR}_i := \frac{|\mathbb{E}[\hat{n}_i \mid \tilde{x}^*]|}{\sqrt{\text{Var}(\hat{n}_i \mid \tilde{x}^*)}} \approx \sqrt{\frac{2N}{\pi}} \frac{s_i(\tilde{x}^*)}{\|\nabla\phi(\tilde{x}^*)\|_2}. \quad (20)$$

*Consequently:*

*(1) Low-sensitivity dimensions lack systematic drift and their perturbations are dominated by initialization and noise.* *If $s_i(\tilde{x}^*)/\|\nabla\phi(\tilde{x}^*)\|_2 \ll \sqrt{\pi/(2N)}$ (equivalently $\text{SNR}_i \ll 1$), then the update in coordinate $i$ is noise-dominated, implying that the expected refinement signal is negligible and the coordinate remains largely determined by the initialization $d_i^{(0)}$.*

*(2) Only $O(N)$ dimensions can be reliably optimized per iteration.* *Let $\mathcal{S} := \{i : \text{SNR}_i \geq 1\}$ be the set of dimensions that can be reliably updated (i.e., with non-negligible expected drift). Then*

$$|\mathcal{S}| \leq \frac{2N}{\pi}, \quad (21)$$

*i.e., at most $O(N)$ dimensions can have non-negligible, reliable refinement signal at the $t$-th iteration.*

**Summary of Proposition 4.2.** This proposition is proved in Appendix A.2. This proposition quantifies the per-coordinate signal-to-noise ratio (SNR) of zeroth-order normal estimation in decision-based attacks, revealing **two** fundamental limitations. First, it shows that the SNR for each dimension $i$ is approximately proportional to the normalized pixel sensitivity $s_i(\tilde{x}^*)/\|\nabla\phi(\tilde{x}^*)\|_2$ and scales with $\sqrt{N}$, where $N$ denotes the number of randomly sampled directions used in the estimation process. When this SNR is far below 1, the corresponding dimension is dominated by noise. Hence its refinement is driven primarily by the initial perturbation rather than by systematic optimization.

Second, Proposition 4.2 establishes an upper bound of $O(N)$ on the number of dimensions that can be reliably refined per iteration (those with SNR $\geq 1$). These results hold under the same local linearity and uniform sampling assumptions as Lemma 4.1, and they provide a theoretical explanation for why decision-based attacks require many queries to achieve low $\ell_2$ distortion: each iteration can effectively refine only a small subset of dimensions, while most other dimensions either remain unchanged or exhibit random-walk–like behavior. This dimension-efficiency bottleneck exposes a fundamental limitation of isotropic refinement and motivates sensitivity-aware strategies that focus refinement on informative, high-impact dimensions rather than treating all coordinates uniformly.

Proposition 4.2 reveals a phenomenon that is not specific to the MC normal estimator, but instead reflects an intrinsic information bottleneck of hard-label black-box attacks. In the decision-based setting, each query yields only a single bit of information, indicating the specific side of the decision boundary on which the queried point lies. As a result, the per-iteration information budget is fundamentally limited, especially in high-dimensional input spaces with heterogeneous coordinate sensitivities.

Under a local linear sign model, this one-bit constraint induces an *effective dimensionality* limit: for any update rule that uses $N$ hard-label queries at an iteration to decide a perturbation increment $\Delta d$, only $O(N)$ input coordinates can exhibit a non-negligible expected drift (equivalently, a coordinate-wise signal-to-noise ratio above a constant threshold), while the remaining coordinates are dominated by stochastic fluctuations. Proposition 4.2 provides a solid theoretical clue of this phenomenon in the context of the MC normal estimator. Establishing a formal minimax lower bound for general adaptive decision-based attacks remains a promising direction for future work.

**Theorem 4.3** (Stationary Alignment and Sensitivity Encoding of MC normal-based Attacks). *Consider the MC normal-based attack defined by the direction update in Eq. (13). Then any perturbation direction $d^*$ at which the MC normal-based update is directionally stationary (i.e., exhibits no expected rotation) satisfies the following properties:*

*(i) Stationary alignment in expectation.* *The expected estimated normal is collinear with $d^*$:*

$$\text{Proj}_{d^*}^{\perp} \mathbb{E}[\hat{n}^* \mid d^*] = 0, \quad (22)$$

*or equivalently, there exists a scalar $\lambda \in \mathbb{R}$ such that*

$$\mathbb{E}[\hat{n}^* \mid d^*] = \lambda d^*. \quad (23)$$

*(ii) Sensitivity encoding up to scale.* *The stationary perturbation direction $d^*$ implicitly encodes the relative importance of different input coordinates. Specifically, for $\forall i = 1, \ldots, D$, the magnitude of each perturbation component satisfies*

$$|d_i^*| \propto s_i(\tilde{x}^*(d^*)) = |\nabla_i\phi(\tilde{x}^*(d^*))|. \quad (24)$$

*(iii) Sensitivity ranking and information gain.* *Consequently, the coordinate-wise ranking induced by $|d_i^*|$ coincides with the ranking by the local sensitivities $s_i(\tilde{x}^*(d^*))$, providing a query-free mechanism to infer sensitivity.*

**Summary of Theorem 4.3.** This theorem is proved in Appendix A.3. At stationarity, MC normal-based attacks

exhibit two structural properties: (i) the expected update aligns with the perturbation direction $d^*$, and (ii) the magnitudes $|d_i^*|$ reflect the relative local sensitivities of the input dimensions, up to a scaling factor. As a result, the final perturbation provides a query-free sensitivity ranking.

**Directional Stationarity Does Not Imply $\ell_2$ Optimality.**
A number of well-known attack methods, including (Carlini & Wagner, 2017; Madry et al., 2018; Chen et al., 2020a), have shown, either explicitly or implicitly, that in realistic settings with curved decision boundaries and noisy gradient or normal estimation, convergence to a stationary point does not guarantee that the resulting adversarial example is $\ell_2$-optimal. This observation reveals a fundamental limitation that goes beyond estimation noise: *stationarity alone is insufficient to guarantee minimum $\ell_2$ distortion*. Motivated by this gap, we introduce a Sensitivity-Aware Rescaling (SAR) algorithm that post-processes the stationary perturbation by redistributing its magnitude according to the sensitivity ranking derived in Theorem 4.3, thereby further reducing the $\ell_2$ norm without requiring additional queries.

---

**Algorithm 1** SAR: Sensitivity-Aware Rescaling
---
1: **Input:** Source image $x$; current best boundary point $\tilde{x}_{\text{best}}^*$ (thus $d_{\text{best}} = \tilde{x}_{\text{best}}^* - x$); indicator $I(\cdot)$; pooling factor $r$; rescaling factor $s = 0.1$; query budget $Q_{\text{SAR}}$.
2: **Output:** Refined perturbation $d_{\text{best}}$.
3: **Initialization:** Initialize a tree-based percentile schedule $\{p_i\}_{i \geq 1}$ over $(0, 1)$ following the dyadic traversal $\{0.5, 0.25, 0.75, 0.125, 0.375, \ldots\}$; iteration $i \leftarrow 1$.
4: **Construct importance map** $I_{\text{imp}} \leftarrow 0 \in \mathbb{R}^{H \times W}$.
5: **for** $(u, w)$ over all spatial locations **do**
6: $\quad I_{\text{imp}}[u, w] \leftarrow \frac{1}{C} \sum_{c=1}^{C} |d_{\text{best}}[c, u, w]|$.
7: **end for**
8: $I_{\text{imp}} \leftarrow \text{Upsample}(\text{AvgPool}(I_{\text{imp}}; r), r)$.
9: **while** Current number of model queries $< Q_{\text{SAR}}$ **do**
10: $\quad p \leftarrow p_i$; $d_{\text{temp}} \leftarrow d_{\text{best}}$; $i \leftarrow i + 1$.
11: $\quad$ Choose threshold $\theta_p$ such that a fraction $p$ of entries in $I_{\text{imp}}$ are $\leq \theta_p$.
12: $\quad M[u, w] \leftarrow \mathbf{1}(I_{\text{imp}}[u, w] \leq \theta_p)$ for all $(u, w)$.
13: $\quad d_{\text{temp}} \leftarrow d_{\text{temp}} \odot \left(1 - sM\right)$.
14: $\quad d_{\text{temp}} \leftarrow d_{\text{temp}} \cdot (\|d_{\text{best}}\|_2 / \|d_{\text{temp}}\|_2)$. (renormalize)
15: $\quad$ **if** $I(x + d_{\text{temp}}) = 1$ **then**
16: $\quad\quad \tilde{x} \leftarrow \tilde{x}^*(d_{\text{temp}})$ $\quad\quad$ (boundary projection)
17: $\quad\quad d_{\text{temp}} \leftarrow \tilde{x} - x$.
18: $\quad\quad$ **if** $\|d_{\text{temp}}\|_2 < \|d_{\text{best}}\|_2$ **then**
19: $\quad\quad\quad d_{\text{best}} \leftarrow d_{\text{temp}}$.
20: $\quad\quad$ **end if**
21: $\quad$ **end if**
22: **end while**
23: **Return** $d_{\text{best}}$
---

## 5. Sensitivity-Aware Rescaling

In Theorem 4.3, we show that at directional stationarity, $|d_i^*|$ induces a ranking consistent with local sensitivity, up to a global scale. Driven by this sensitivity-encoding property, we aim to post-process the current boundary perturbation by preferentially shrinking perturbations on low-sensitivity coordinates to further reduce the $\ell_2$ norm.

However, in decision-based black-box attacks, the query budget is severely limited and the hard-label feedback is inherently noisy. As formalized in Proposition 4.2, the raw per-pixel magnitudes in $|d^*|$ can be unreliable, particularly on low-sensitivity coordinates dominated by MC noise and initialization effects. To obtain a more robust estimate, we construct a denoised importance map $Imp$ by aggregating $|d^*|$ across the three color channels, followed by 2D average pooling with factor $r$ over local spatial blocks and subsequent upsampling to the original resolution. This spatial smoothing enforces local consistency and suppresses isolated noise artifacts, yielding a stable importance ranking. Further details are provided in Appendix B.

Based on $Imp$, we define a rescaling operator $\text{Rescal}(p, s)$, which suppresses the perturbations on the $p$ least-important pixels by a rate $s \in [0, 1)$, i.e., the selected pixels are multiplied by $(1 - s)$. At iteration $i$, we apply $\text{Rescal}(p_i, s)$ to the current perturbation, project the rescaled perturbation back to the decision boundary, and query the model to determine whether the new adversarial example has a smaller $\ell_2$ boundary distance. We accept the update only if the distance is reduced; otherwise, we retain the previous perturbation.

In practice, the sensitivity ranking in Theorem 4.3 typically remains valid after applying $\text{Rescal}(p, s)$, since SAR is activated only after the base attack has entered a stable regime. At this stage, the adversarial example is already close to the original image, the perturbation is small, and the local boundary geometry is sufficiently regular for reliable reprojection. Consequently, the sensitivity ranking remains effective in real-world decision-based attacks.

To systematically explore different denoising levels, we schedule $p_i \in (0, 1)$ to follow a tree-based, coarse-to-fine traversal (e.g., $0.5, 0.25, 0.75, 0.125, \ldots$). This strategy yields near-uniform coverage of $(0, 1)$ while progressively refining perturbations across pixels with varying importance. The procedure stops once the query budget is exhausted, and the best perturbation found so far is returned. A detailed line-by-line explanation of the SAR algorithm, as presented in Algorithm 1, is provided below.

**Line 1** of Algorithm 1 specifies the inputs: the source image $x$, the current best boundary point $\tilde{x}_{\text{best}}^*$ which is optimized by the base attacker (e.g., HSJA), the hard-label indicator $I(\cdot)$ (defined in Eq. (2)), the pooling factor $r$ for spatial smoothing, the rescaling factor $s = 0.1$, and the SAR query

budget $Q_{\text{SAR}}$. **Line 3** initializes iteration number $i \leftarrow 1$ and precomputes a coarse-to-fine percentile schedule $\{p_i\}_{i \geq 1}$ over $(0, 1)$ based on dyadic fractions. This schedule enables progressive refinement without committing to a single sensitivity cutoff. **Lines 4-8** construct the importance map $I_{\text{imp}} \in \mathbb{R}^{H \times W}$. **Lines 5-6** compute a pixel-level proxy by channel-aggregating the magnitude of the current best perturbation. This aggregation transforms the coordinate-wise magnitude pattern (Theorem 4.3) into a spatial importance map, aligning with the common interpretation of sensitivity in images. **Line 8** smooths the map via average pooling and upsampling, promoting spatial coherence and reducing isolated noisy artifacts. **Line 9** starts the main loop. **Line 10** sets the current percentile $p \leftarrow p_i$, i.e., the fraction of pixels to be treated as low-importance pixels at current iteration $i$. Then copy the current best perturbation $d_{\text{temp}} \leftarrow d_{\text{best}}$ as an attempt perturbation. **Lines 11-14** detailed the rescaling operator $\text{Rescal}(p, s)$. **Line 11** defines the cutoff $\theta_p$ as the lower $p$ percentile threshold of $I_{\text{imp}}$: a fraction $p$ of spatial entries that satisfy $I_{\text{imp}}[u, w] \leq \theta_p$. **Line 12** constructs the binary mask $M[u, w] = \mathbf{1}(I_{\text{imp}}[u, w] \leq \theta_p)$ to select the top-$p$ spatial locations with the smallest importance values. **Line 13** applies sensitivity-aware rescaling where $M$ is broadcast across all $C$ channels. This operation downweights coordinates inside the selected region by a factor $s = 0.1$, i.e., it multiplies them by $1 - s = 0.9$, thereby mitigating the low-sensitivity stagnation effect. **Line 14** renormalizes $d$ to match the reference norm $\|d_{\text{best}}\|_2$. Together with the small rescaling factor $s$, this enables cumulative small-step redistribution without inflating the overall $\ell_2$ scale. **Line 15** checks feasibility using the hard-label oracle: the boundary projection is performed only if the perturbed point $x + d_{\text{temp}}$ remains adversarial. **Line 16** projects the candidate along direction $d_{\text{temp}}$ back onto the decision boundary by computing the corresponding boundary point $\tilde{x} \leftarrow \tilde{x}^*(d_{\text{temp}})$ (defined in Eq. (6); implemented in practice by binary search). **Line 17** updates the boundary perturbation as $d_{\text{temp}} \leftarrow \tilde{x} - x$. **Lines 18-19** accept the update only if it reduces the $\ell_2$ distance. **Line 23** returns the refined perturbation $d_{\text{best}}$.

# 6. Experiments

In this section, we evaluate the proposed SAR algorithm and validate the theoretical insights developed in Section 4. SAR does not impose any change to the original attacker pipeline. Instead, after a base decision-based attack terminates, SAR estimates a sensitivity profile from the obtained adversarial example and further refines the per-pixel perturbation magnitudes to reduce the overall $\ell_2$ distortion.

We conduct experiments on three benchmark datasets, ImageNet, MNIST, and CIFAR-10, using common architectures including ViT, VGG, ResNet, and Wide ResNet

(WRN), which are widely used in state-of-the-art decision-based attacks (Chen & Gu, 2020; Chen et al., 2020a; Wang et al., 2025a). We further test SAR under representative defense settings, such as adversarial training and Lipschitz/robustness-regularized models (Araujo et al., 2023; Tsuzuku et al., 2018). Performance is evaluated by average/median $\ell_2$ distortion, attack success rate (ASR), and a perceptual quality metric called *Structural Similarity Index* (SSIM) (Wang et al., 2004).

## 6.1. Experimental Settings

**Competing approaches.** We compare SAR against three representative decision-based attacks: HSJA (Chen et al., 2020a), CGBA (Reza et al., 2023), and TtBA (Wang et al., 2025a). All three are state-of-the-art ZO normal-estimation methods, with TtBA serving as a strong recent baseline that achieves cutting-edge $\ell_2$ performance. We evaluate SAR under 6 configurations: the three base attackers (HSJA, CGBA, and TtBA) and their corresponding "Attacker + SAR" variants. SSIM is a perceptual metric that aligns more closely with human visual perception than pixel-wise distance measures; higher SSIM values indicate perturbations that are less perceptible to human observers. We compute SSIM between the clean and adversarial images (typically on the luminance channel with a standard local-window aggregation), and report the average across successful attacks.

**Benchmark datasets and models.** On ImageNet (Deng et al., 2009), we evaluate standard architectures including VGG19 (Simonyan & Zisserman, 2015), ResNet50 (He et al., 2016), Inception-v3 (Szegedy et al., 2016), and the Vision Transformer (ViT) (Dosovitskiy et al., 2021). To assess robustness under defenses, we additionally include an adversarially trained ImageNet model from the MadryLab Robustness library, denoted as Engstrom (Engstrom et al., 2019). On CIFAR-10 (Krizhevsky et al., 2009), we evaluate a standard CNN model from (Chen & Gu, 2020) and an adversarially trained WideResNet (WRN) from (Wang et al., 2023). On MNIST (LeCun et al., 1998), we use a CNN model from (Chen & Gu, 2020) and a robust Lipschitz-constrained model (LMT) (Tsuzuku et al., 2018).

For each model, we randomly sample 500 images from the test set and run each attack under a fixed query budget of 10,000 queries. We define an attack as successful if its $\ell_2$ perturbation norm is at most $\epsilon = 3.0$ on ImageNet and MNIST, and $\epsilon = 1.0$ on CIFAR-10.

**Hyperparameter settings.** We adopt the recommended hyperparameter settings from (Chen et al., 2020a; Reza et al., 2023; Wang et al., 2025a). Specifically, for SAR, the decision-boundary search tolerance is set to $10^{-4}$. Based on the parameter sensitivity analysis in Appendix B, we set the SAR query budget to $Q_{\text{SAR}} = 300$, the base attacker query budget to $Q_{\text{BASE}} = 9700$, and the rescaling factor to

*Table 1.* Average $\ell_2$ distortion across different attack settings ($\downarrow$: +SAR improves over the corresponding base attack).

| Dataset | ImageNet | | | | | CIFAR-10 | | MNIST | |
|---|---|---|---|---|---|---|---|---|---|
| Model | ViT | ResNet | Inception | VGG | Engstrom | CNN | WRN | CNN | LMT |
| HSJA | 2.9681 | 8.0975 | 4.1391 | 2.4745 | 11.260 | 0.4517 | 3.2171 | 2.8121 | 3.9416 |
| HSJA+SAR | 2.5712↓ | 7.9959↓ | 4.2420 | 2.0545↓ | 10.292↓ | 0.4395↓ | 3.1207↓ | 2.6499↓ | 3.5914↓ |
| CGBA | 1.5685 | 6.6303 | 4.5076 | 1.1743 | 7.4202 | 0.1797 | 1.3444 | 1.4872 | 3.6511 |
| CGBA+SAR | **1.3542↓** | 3.9326↓ | 2.4048↓ | **0.9596↓** | 6.5115↓ | 0.1727↓ | 1.2719↓ | 1.3032↓ | 2.6303↓ |
| TtBA | 1.5019 | 3.0627 | 1.9609 | 1.1498 | 6.4504 | 0.1734 | 1.2764 | 1.3019 | 2.0461 |
| TtBA+SAR | 1.3613↓ | **2.6883↓** | **1.8411↓** | 0.9829↓ | **5.4437↓** | **0.1682↓** | **1.2088↓** | **1.2852↓** | **1.9725↓** |

*Table 2.* ASR across different attack settings ($\uparrow$: +SAR improves over the corresponding base attack).

| Dataset | ImageNet | | | | | CIFAR-10 | | MNIST | |
|---|---|---|---|---|---|---|---|---|---|
| Model | ViT | ResNet | Inception | VGG | Engstrom | CNN | WRN | CNN | LMT |
| HSJA | 62.6% | 34.2% | 58.8% | 72.2% | 17.6% | 92.0% | 15.6% | 73.8% | 20.0% |
| HSJA+SAR | 71.8%↑ | 41.4%↑ | 61.6%↑ | 79.6%↑ | 26.4%↑ | 93.2%↑ | 23.8%↑ | 84.4%↑ | 33.8%↑ |
| CGBA | 84.8% | 58.0% | 76.2% | 93.0% | 30.6% | 98.8% | 36.2% | 98.0% | 76.2% |
| CGBA+SAR | 90.4%↑ | 58.8%↑ | 79.8%↑ | **95.8%↑** | 38.0%↑ | 99.4%↑ | 42.0%↑ | 98.8%↑ | 82.2%↑ |
| TtBA | 86.0% | 67.8% | 81.4% | 93.0% | 43.2% | 99.2% | 37.2% | 99.0% | 87.4% |
| TtBA+SAR | **90.6%↑** | **69.4%↑** | **84.6%↑** | 94.4%↑ | **46.8%↑** | **99.8%↑** | **44.8%↑** | **99.4%↑** | **90.6%↑** |

*Table 3.* Performance on the ImageNet dataset using a ViT model

| Attackers | ASR | AVG $\ell_2$ | MED $\ell_2$ | SSIM |
|---|---|---|---|---|
| HSJA | 62.6% | 2.9681 | 2.2590 | 0.9758 |
| HSJA+PAR | 70.0% | 2.7396 | 2.1388 | 0.9793 |
| HSJA+SAR | 71.8% | 2.5712 | 1.8941 | 0.9813 |
| CGBA | 84.8% | 1.5685 | 1.1233 | 0.9937 |
| CGBA+PAR | 89.2% | 1.4584 | 1.1031 | 0.9950 |
| CGBA+SAR | 90.4% | **1.3542** | 1.0382 | 0.9954 |
| TtBA | 86.0% | 1.5019 | 0.9969 | 0.9949 |
| TtBA+PAR | 88.2% | 1.4793 | 0.9967 | 0.9949 |
| TtBA+SAR | **90.6%** | 1.3613 | **0.9493** | **0.9955** |

*Table 4.* Performance on the ImageNet dataset using a VGG model

| Attackers | ASR | AVG $\ell_2$ | MED $\ell_2$ | SSIM |
|---|---|---|---|---|
| HSJA | 72.2% | 2.4745 | 1.6251 | 0.9838 |
| HSJA+PAR | 79.6% | 2.0897 | 1.3951 | 0.9870 |
| HSJA+SAR | 79.6% | 2.0545 | 1.2663 | 0.9874 |
| CGBA | 93.0% | 1.1743 | 0.7503 | 0.9964 |
| CGBA+PAR | 94.0% | 1.0406 | 0.6475 | 0.9971 |
| CGBA+SAR | **95.8%** | **0.9596** | **0.5733** | **0.9973** |
| TtBA | 93.0% | 1.1498 | 0.7351 | 0.9966 |
| TtBA+PAR | 93.2% | 1.0685 | 0.7096 | 0.9969 |
| TtBA+SAR | 94.4% | 0.9829 | 0.6137 | 0.9972 |

$s = 0.1$. We set the average pooling reduction factor $r$ to 16 for ImageNet, 4 for CIFAR-10, and 7 for MNIST. All experiments were run on an Intel Xeon Gold 6330 CPU with four NVIDIA GeForce RTX 4090 GPUs using PyTorch 2.3.0, Torchvision 0.18.0, and Python 3.11.5.

## 6.2. Experimental Results

**Main results.** As shown in Tables 1 and 7 (in Appendix C), the *CGBA + SAR* and *TtBA + SAR* variants achieve the best performance in terms of both average and median $\ell_2$ norms across all datasets and target models. Compared with the corresponding base attackers alone, under the same total query budget, *Attacker + SAR* reduces the average and median $\ell_2$ distortion by approximately 10%. Notably, SAR remains effective for robust models, including Engstrom, WRN, and LMT, consistently yielding lower $\ell_2$ norms.

In addition to reducing perturbation magnitude, SAR also improves the overall attack success rate (ASR). As reported

in Table 2, *TtBA + SAR* achieves the highest ASR on 8 out of 9 evaluated models. The only exception is the VGG model, on which *CGBA + SAR* attains the highest ASR of 95.8%.

Furthermore, SAR effectively reduces the perturbation perceptual visibility. According to the SSIM results in Table 8 (in Appendix C), *TtBA + SAR* achieves the highest SSIM on 8 out of 9 models, while *CGBA + SAR* attains the highest SSIM on the VGG model. These results indicate that SAR reduces the perceptual visibility of adversarial examples, enhancing their practical usefulness in real-world settings.

Overall, the *Attacker + SAR* variants consistently achieve superior performance across all evaluation metrics on both standard and adversarially defended models. For any given base attacker (e.g., HSJA), incorporating SAR consistently outperforms the attacker alone in terms of average and median $\ell_2$ norms, ASR, and SSIM. Notably, these improvements are obtained by reallocating only 3% (300 out of 10,000) of the total query budget from the base attacker to

Table 5. Average $\ell_2$ distortion under 5,000 query budget.

| Dataset | ImageNet | | CIFAR-10 | MNIST |
|---|---|---|---|---|
| Model | ViT | VGG | CNN | CNN |
| HSJA | 5.377 | 4.207 | 0.738 | 3.656 |
| HSJA+SAR | 5.238 | 4.169 | 0.721 | 3.429 |
| CGBA | 2.408 | 1.690 | 0.214 | 1.705 |
| CGBA+SAR | **2.010** | **1.499** | 0.199 | 1.579 |
| TtBA | 2.193 | 1.765 | 0.225 | 2.272 |
| TtBA+SAR | 2.070 | 1.561 | **0.198** | **1.356** |

SAR, without increasing the overall query budget. These results validate the sensitivity ranking theorem (Theorem 4.3), confirming that sensitivity-aware rescaling can significantly improve the effectiveness of decision-based ZO normal-estimation attacks.

**Comparative experiments with other region-sensitive methods.** Under strict decision-based settings, applicable region-sensitive methods are highly limited, since most existing approaches rely on gradient or score information. PAR (Shi et al., 2022) is one of the few methods that satisfies this constraint while incorporating region-level sensitivity modeling, making it a suitable baseline. PAR makes binary decisions for each region by either preserving or removing its perturbation, resulting in a coarse all-or-nothing strategy that contrasts with the fine-grained rescaling mechanism of SAR. The results in Table 3 and Table 4 show that PAR achieves limited performance because it ignores the intrinsic structure of perturbations and relies on coarse binary updates. In contrast, SAR captures fine-grained sensitivity patterns and performs continuous perturbation rescaling, enabling more precise and effective refinement. As a result, SAR consistently achieves better performance.

**Performance under 5,000 Total Query Budget.** SAR builds on a base decision-based attack that has produced a reasonably accurate boundary point, such that the resulting perturbation encodes meaningful sensitivity information. When the base attack is less well converged (e.g., under very low query budgets with unstable perturbations), the inferred importance map becomes less reliable, potentially resulting in reduced effectiveness. However, experimental results in Table 5 provide strong evidence that SAR remains highly effective in the practically relevant regime where the base attack is not yet fully converged but has already captured meaningful partial sensitivity information. Specifically, even with a query budget of 5,000, where the base attack is noticeably less converged than under 10,000 queries, SAR can manage to deliver substantial and consistent reductions in the $\ell_2$ norm.

This robustness primarily arises from the average pooling and upsampling steps in SAR, which make the inferred importance map more resilient to noise. Even before full

convergence, the perturbation produced by the base attack often already contains coarse but useful sensitivity structure, which allows SAR to remain beneficial.

## 7. Conclusion and Future Work

This work provides a unified theoretical account of efficiency and convergence phenomena in hard-label (decision-based) attacks. We first characterize the statistical structure of directional probing and normal estimation near the decision boundary (Lemma 4.1), and demonstrate how one-bit feedback constrains the amount of usable information per query, effectively limiting reliable refinement to only a small subset of coordinates (Proposition 4.2). We then show that stationary hard-label-driven updates align (in expectation) with the decision-boundary normal and that their coordinate magnitudes capture local sensitivity (Theorem 4.3). Guided by Theorem 4.3, we develop *Sensitivity-Aware Rescaling* (SAR), a query-efficient refinement algorithm that leverages the sensitivity ranking embedded in the current perturbation.

Our study marks a principled shift from isotropic sampling to *sensitivity-adaptive* decision attacks: since Proposition 4.2 exposes a one-bit efficiency bottleneck, future algorithms should concentrate queries on high-SNR (high-sensitivity) dimensions and dynamically adjust their sampling distributions. A natural direction for future work is therefore to move beyond normal estimation, developing estimator-agnostic analyses and oracle-level characterizations, as well as minimax lower bounds that jointly capture the one-bit bottleneck and heterogeneous sensitivity.

## Acknowledgments

This research is supported by the China Scholarship Council (CSC, Grant No. 202506470044), the BUPT Excellent Ph.D. Students Foundation (Grant No. CX20241003), and the National Research Foundation, Singapore and Infocomm Media Development Authority under its Trust Tech Funding Initiative. Any opinions, findings and conclusions or recommendations expressed in this material are those of the author(s) and do not reflect the views of the National Research Foundation, Singapore and Infocomm Media Development Authority. This research is supported by A*STAR Career Development Fund <Project No. C243512010>.

## Impact Statement

This paper advances AI security research by improving the understanding of black-box (hard-label) adversarial attacks. We recognize that adversarial attack techniques can be dual-use: while they are useful for robustness assessment and defense design, they could also be misused to undermine deployed systems. Our method is presented for research and

benchmarking purposes, and the insights may generalize to other modalities and tasks such as speech recognition and text classification. Overall, we expect this work to contribute to safer and more trustworthy AI as such technologies are increasingly adopted in real-world applications.

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

# A. Proof

## A.1. Proof of Lemma 4.1

*Proof.* **Step 1 (Normalization is w.l.o.g.).** Assume $\nabla\phi(\tilde{x}^*) \neq 0$. Using $\mathrm{sign}(ct) = \mathrm{sign}(t)$ for $c > 0$,

$$\mathrm{sign}\big(\nabla\phi(\tilde{x}^*)^\top u\big) = \mathrm{sign}\left(\left(\frac{\nabla\phi(\tilde{x}^*)}{\|\nabla\phi(\tilde{x}^*)\|_2}\right)^\top u\right). \tag{25}$$

For convenience, denote

$$v := \frac{\nabla\phi(\tilde{x}^*)}{\|\nabla\phi(\tilde{x}^*)\|_2} \in \mathbb{S}^{D-1}. \tag{26}$$

**Step 2 (Single-sample reduction).** Under the local sign model, define for $u \sim \mathrm{Unif}(\mathbb{S}^{D-1})$

$$Z_i := \mathrm{sign}(v^\top u)\, u_i. \tag{27}$$

Then each MC component is a sample mean:

$$\hat{n}_i \approx \frac{1}{N} \sum_{k=1}^{N} Z_{k,i},\ Z_{k,i} := \mathrm{sign}(v^\top u_k)\,(u_k)_i. \tag{28}$$

**Step 3 (Mean via rotational symmetry).** Let

$$m := \mathbb{E}\big[\mathrm{sign}(v^\top u)\, u\big] \in \mathbb{R}^D. \tag{29}$$

By rotational invariance of $u \sim \mathrm{Unif}(\mathbb{S}^{D-1})$, $m$ must be colinear with $v$, so $m = \beta_D v$ for some scalar $\beta_D$. Projecting onto $v$ yields

$$\beta_D = v^\top m = \mathbb{E}\big[\mathrm{sign}(v^\top u)\, v^\top u\big] = \mathbb{E}[|v^\top u|] =: c_D. \tag{30}$$

where we used $\mathrm{sign}(t)\, t = |t|$. Therefore, $\mathbb{E}[Z_i] = (m)_i = c_D v_i$, and by Eq. (28),

$$\mathbb{E}[\hat{n}_i] \approx c_D v_i = c_D\, \frac{\nabla_i\phi(\tilde{x}^*)}{\|\nabla\phi(\tilde{x}^*)\|_2}, \tag{31}$$

which is Eq. (15).

**Step 4 (Variance).** Since $\{Z_{k,i}\}_{k=1}^{N}$ are i.i.d.,

$$\mathrm{Var}(\hat{n}_i) \approx \frac{1}{N}\mathrm{Var}(Z_i). \tag{32}$$

Moreover, $\mathrm{sign}(\cdot)^2 = 1$ implies $Z_i^2 = u_i^2$, hence

$$\mathbb{E}[Z_i^2] = \mathbb{E}[u_i^2]. \tag{33}$$

For $u \sim \mathrm{Unif}(\mathbb{S}^{D-1})$, symmetry gives $\mathbb{E}[u_1^2] = \cdots = \mathbb{E}[u_D^2]$, and since $\sum_{j=1}^{D} u_j^2 = 1$ a.s., taking expectation yields

$$\mathbb{E}[u_i^2] = \frac{1}{D}. \tag{34}$$

Combining Eq. (32)–Eq. (34) and $\mathbb{E}[Z_i] = c_D v_i$ gives

$$\mathrm{Var}(\hat{n}_i) \approx \frac{1}{N}\left(\frac{1}{D} - (c_D v_i)^2\right) = \frac{1}{N}\left(\frac{1}{D} - c_D^2\, \frac{\nabla_i\phi(\tilde{x}^*)^2}{\|\nabla\phi(\tilde{x}^*)\|_2^2}\right). \tag{35}$$

which is Eq. (16).

**Step 5 (Closed form and asymptotics of $c_D$).** By rotational invariance we may take $v = e_1$ and set $T := v^\top u = u_1$. Then $T$ has density on $[-1, 1]$

$$f_T(t) = C_D(1 - t^2)^{\frac{D-3}{2}}, \quad C_D = \frac{\Gamma(\frac{D}{2})}{\sqrt{\pi}\Gamma(\frac{D-1}{2})}, \tag{36}$$

where $C_D$ is the normalizing constant. Therefore,

$$c_D = \mathbb{E}[|T|] = 2\int_0^1 t f_T(t)\,dt = 2C_D \int_0^1 t(1 - t^2)^{\frac{D-3}{2}}\,dt. \tag{37}$$

Let $s = t^2$ (so $ds = 2t\,dt$). Then

$$\int_0^1 t(1 - t^2)^{\frac{D-3}{2}}\,dt = \frac{1}{2}\int_0^1 (1 - s)^{\frac{D-3}{2}}\,ds = \frac{1}{D - 1}. \tag{38}$$

Substituting Eq. (38) into Eq. (37) gives

$$c_D = \frac{2C_D}{D - 1} = \frac{\Gamma(\frac{D}{2})}{\sqrt{\pi}\Gamma(\frac{D+1}{2})}, \tag{39}$$

where we used $\Gamma(\frac{D+1}{2}) = \frac{D-1}{2}\Gamma(\frac{D-1}{2})$. This matches Eq. (17). And then, for large $D$, letting $z := D/2$, we have $c_D = \Gamma(z)/(\sqrt{\pi}\Gamma(z + \frac{1}{2}))$, and the standard Gamma-ratio asymptotic $\Gamma(z + a)/\Gamma(z + b) \sim z^{a-b}$ as $z \to \infty$ yields

$$c_D \sim \frac{1}{\sqrt{\pi}} z^{-1/2} = \sqrt{\frac{2}{\pi D}}, \tag{40}$$

which implies Eq. (18). $\qquad\square$

## A.2. Proof of Proposition 4.2

*Proof.* Fix the $t$-th iteration at the current boundary point $\tilde{x}^*$. Assume $\nabla\phi(\tilde{x}^*) \neq 0$ and recall $s_i(\tilde{x}^*) := |\nabla_i\phi(\tilde{x}^*)|$.

**Step 1 (SNR expression).** By Lemma 4.1, with $c_D \approx \sqrt{2/(\pi D)}$ for large $D$,

$$\mathrm{SNR}_i := \frac{|\mathbb{E}[\hat{n}_i \mid \tilde{x}^*]|}{\sqrt{\mathrm{Var}(\hat{n}_i \mid \tilde{x}^*)}} \approx \frac{c_D |\nabla_i\phi(\tilde{x}^*)|/\|\nabla\phi(\tilde{x}^*)\|_2}{\sqrt{\frac{1}{N}\left(\frac{1}{D} - c_D^2\frac{\nabla_i\phi(\tilde{x}^*)^2}{\|\nabla\phi(\tilde{x}^*)\|_2^2}\right)}} \approx c_D\sqrt{ND} \cdot \frac{|\nabla_i\phi(\tilde{x}^*)|}{\|\nabla\phi(\tilde{x}^*)\|_2} \approx \sqrt{\frac{2N}{\pi}} \cdot \frac{s_i(\tilde{x}^*)}{\|\nabla\phi(\tilde{x}^*)\|_2}. \tag{41}$$

This is Eq. (20). The middle approximation uses that, in the low-sensitivity regime, the variance term is dominated by the $1/(ND)$ floor.

**Step 2 (Conclusion (1): initialization-dominated when $\mathrm{SNR}_i \ll 1$).** The coordinate-wise update increment is Conditioned on $\tilde{x}^*$,

$$\mathbb{E}[\Delta d_i^{(t)} \mid \tilde{x}^*] = \alpha_t\,\mathbb{E}[\hat{n}_i \mid \tilde{x}^*],$$
$$\mathrm{Var}(\Delta d_i^{(t)} \mid \tilde{x}^*) = \alpha_t^2\,\mathrm{Var}(\hat{n}_i \mid \tilde{x}^*). \tag{42}$$

Therefore, the SNR of the *increment* equals $\mathrm{SNR}_i$ (the factor $\alpha_t$ cancels). If $\mathrm{SNR}_i \ll 1$ (equivalently $s_i(\tilde{x}^*)/\|\nabla\phi(\tilde{x}^*)\|_2 \ll \sqrt{\pi/(2N)}$ from Eq. (20)), then the update in coordinate $i$ is noise-dominated, i.e., the systematic drift is much smaller than its stochastic fluctuation. Consequently, coordinate $i$ cannot be reliably steered by $\hat{n}$ and its refined value remains largely determined by the existing value carried from earlier iterations, ultimately by the initialization $d_i^{(0)}$. This proves statement (1).

**Step 3 (Conclusion (2): $|\mathcal{S}| \leq 2N/\pi$).** Define

$$r_i := \frac{s_i(\tilde{x}^*)}{\|\nabla\phi(\tilde{x}^*)\|_2} = \frac{|\nabla_i\phi(\tilde{x}^*)|}{\|\nabla\phi(\tilde{x}^*)\|_2}, \text{ and } \sum_{i=1}^{D} r_i^2 = 1. \tag{43}$$

Let $\mathcal{S} := \{i : \mathrm{SNR}_i \geq 1\}$. From Eq. (20), $\mathrm{SNR}_i \geq 1$ implies $r_i \gtrsim \sqrt{\pi/(2N)}$ for all $i \in \mathcal{S}$. Therefore,

$$1 = \sum_{i=1}^{D} r_i^2 \geq \sum_{i\in\mathcal{S}} r_i^2 \gtrsim |\mathcal{S}| \cdot \frac{\pi}{2N}, \tag{44}$$

which yields $|\mathcal{S}| \lesssim 2N/\pi$. At the same approximation level as Eq. (20), we write $|\mathcal{S}| \leq 2N/\pi$, proving Eq. (21). $\qquad\square$

### A.3. Proof of Theorem 4.3

*Proof sketch.* **Part (i): Stationary alignment.** We analyze the long-run behavior of the MC normal-based update. Let

$$u^{(t)} = \frac{d^{(t)}}{\|d^{(t)}\|_2}$$

denote the normalized perturbation direction. The update component orthogonal to $u^{(t)}$ is given by

$$\hat{n}_{\perp}^{(t)} = \mathrm{Proj}_{u^{(t)}}^{\perp} \hat{n}^{(t)}.$$

Under Robbins–Monro step sizes, the evolution of $u^{(t)}$ is governed, in expectation, by the mean drift

$$h(u) = \mathbb{E}[\hat{n}_{\perp} \mid u].$$

At a stationary direction $u^*$, this drift vanishes, i.e., $h(u^*) = 0$, which implies that $\mathbb{E}[\hat{n} \mid u^*]$ has no component orthogonal to $u^*$ and is therefore collinear with $u^*$. Since $d^* = g(d^*)\,u^*$, collinearity with $u^*$ directly implies collinearity with $d^*$, establishing Eq. (22)–(23).

**Part (ii): Sensitivity encoding.** By Lemma 4.1, under the local linear sign model the expected MC normal estimator at a boundary point satisfies

$$\mathbb{E}[\hat{n}_i \mid \tilde{x}^*] = c_D \frac{\nabla_i\phi(\tilde{x}^*)}{\|\nabla\phi(\tilde{x}^*)\|_2}.$$

At a stationary perturbation direction $d^*$, Part (i) implies that there exists a scalar $\lambda \neq 0$ such that

$$\mathbb{E}[\hat{n}_i \mid d^*] = \lambda d_i^*.$$

Combining the two expressions yields

$$\lambda d_i^* = c_D \frac{\nabla_i\phi(\tilde{x}^*(d^*))}{\|\nabla\phi(\tilde{x}^*(d^*))\|_2}.$$

Taking absolute values on both sides establishes Eq. (24).

**Part (iii): Sensitivity ranking.** The ranking result follows immediately from Eq. (24), since proportionality preserves coordinate-wise ordering. $\qquad\square$

## B. Parameter Sensitivity Analysis

Our SAR approach in Algorithm 1 introduces several key hyperparameters, including the rescaling factor $s$, the SAR query budget $Q_{\mathrm{SAR}}$, and the average pooling reduction factor $r$ for different datasets. Unless otherwise specified, we set $s = 0.1$ and $Q_{\mathrm{SAR}} = 300$ out of a total query budget of 10,000. These settings are selected based on preliminary experiments conducted on the ImageNet dataset using the VGG19 model and the TtBA base attacker. Similar trends are consistently observed across other base attackers and target models.

*Table 6.* Additional Hyperparameter Ablations.

*(a)* Impact of $s$ and SAR query budget $Q_{\mathrm{SAR}}$ (ImageNet dataset, VGG model, and TtBA approach).

| $s\backslash Q$ | 200 | 300 | 400 | 500 | 600 | 700 |
|---|---|---|---|---|---|---|
| 0.05 | 1.1050 | 1.0792 | 1.0931 | 1.0994 | 1.1272 | 1.1368 |
| | (0.6580) | (0.6551) | (0.6453) | (0.6497) | (0.6510) | (0.6520) |
| 0.08 | 1.0858 | 1.0657 | 1.0862 | 1.0970 | 1.1170 | 1.1338 |
| | (0.6457) | (0.6437) | (0.6439) | (0.6472) | (0.6493) | (0.6527) |
| 0.10 | 1.0809 | **1.0615** | 1.0859 | 1.0945 | 1.1188 | 1.1334 |
| | (0.6433) | **(0.6428)** | (0.6438) | (0.6465) | (0.6486) | (0.6522) |
| 0.12 | 1.0816 | 1.0636 | 1.0866 | 1.0940 | 1.1193 | 1.1344 |
| | (0.6440) | (0.6443) | (0.6447) | (0.6480) | (0.6500) | (0.6534) |
| 0.15 | 1.0892 | 1.0872 | 1.0868 | 1.0989 | 1.1218 | 1.1336 |
| | (0.6449) | (0.6451) | (0.6453) | (0.6498) | (0.6522) | (0.6545) |

*(b)* Impact of pooling factor $r$ (ImageNet).

| Pooling factor $r$ | 1 (3 channels) | 1 | 2 | 4 | 8 | 16 | 32 |
|---|---|---|---|---|---|---|---|
| VGG+TtBA | 1.43(1.07) | 1.39(1.05) | 1.38(1.03) | 1.36(1.00) | 1.34(0.96) | **1.29(0.94)** | 1.29(0.95) |
| ViT+TtBA | 1.67(1.13) | 1.61(1.12) | 1.60(1.11) | 1.56(1.10) | 1.52(1.10) | **1.51(1.08)** | 1.58(1.09) |
| VGG+HSJA | 3.83(2.63) | 3.82(2.62) | 3.62(2.28) | 3.53(2.19) | 3.47(2.19) | **3.41(2.18)** | 3.43(2.24) |

*(c)* Impact of pooling factor $r$ (CIFAR-10).

| Pooling factor $r$ | 1 | 2 | 4 | 8 |
|---|---|---|---|---|
| WRN+TtBA | 1.26(1.10) | 1.19(1.06) | **1.16(1.01)** | 1.22(1.07) |
| CNN+HSJA | 0.43(0.44) | 0.40(0.41) | **0.38(0.40)** | 0.40(0.42) |

*(d)* Impact of pooling factor $r$ (MNIST).

| Pooling factor $r$ | 1 | 2 | 4 | 7 | 14 |
|---|---|---|---|---|---|
| LMT+CGBA | 1.72(1.74) | 1.70(1.72) | 1.70(1.72) | **1.68(1.70)** | 1.71(1.73) |
| CNN+HSJA | 2.71(2.26) | 2.66(2.17) | 2.43(1.73) | **2.33(1.62)** | 2.70(2.27) |

To assess the impact of these parameters, we conduct a comprehensive sensitivity analysis. Specifically, we vary the rescaling factor $s$ and the SAR query budget $Q_{\mathrm{SAR}}$ (Table 6-(a)), as well as the average pooling reduction factor $r$ for ImageNet (Table 6-(b)), CIFAR-10 (Table 6-(c)), and MNIST (Table 6-(d)). The best configurations are highlighted in **bold**.

As shown in the results, the parameter combination $(s = 0.1, Q_{\mathrm{SAR}} = 300)$ consistently achieves the best performance. For the average pooling reduction factor, we observe optimal values of $r = 16$ for ImageNet, $r = 4$ for CIFAR-10, and $r = 7$ for MNIST. Notably, performance variations around these selected values are relatively small, indicating that SAR is not highly sensitive to moderate changes in these parameters.

For the importance map design, we model importance at the pixel level rather than incorporating channel-wise information. This choice is motivated by the prior that sensitivities across the three color channels are highly correlated, with the channels typically exhibiting uniformly high or low sensitivity at the same spatial location. Therefore, explicitly modeling channel-wise importance is likely to provide only marginal gains while incurring a higher query cost. Empirically, under the same pooling factor $r = 1$, the pixel-wise importance map achieves better $\ell_2$ performance than its channel-wise variant.

## C. Appendix Experimental Results

Table 7 shows that the *CGBA+SAR* and *TtBA+SAR* variants consistently achieve the best performance in terms of both average and median $\ell_2$ norms across all datasets and target models.

Moreover, SAR effectively reduces the perceptual visibility of adversarial perturbations. As shown by the SSIM results in Table 8, *TtBA+SAR* attains the highest SSIM on 8 out of 9 target models, while *CGBA+SAR* achieves the best SSIM on the VGG model. These results indicate that SAR significantly improves the imperceptibility of adversarial examples, enhancing their practicality in real-world scenarios.

*Table 7.* Median $\ell_2$ distortion across different attack settings.

| Dataset | ImageNet | | | | | CIFAR-10 | | MNIST | |
|---|---|---|---|---|---|---|---|---|---|
| Model | ViT | ResNet | Inception | VGG | Engstrom | CNN | WRN | CNN | LMT |
| HSJA | 2.2590 | 5.3864 | 2.3103 | 1.6251 | 8.8542 | 0.4194 | 2.9679 | 2.4530 | 3.8705 |
| HSJA+SAR | 1.8941↓ | 4.9589↓ | 2.2462↓ | 1.2663↓ | 7.2587↓ | 0.4079↓ | 2.8016↓ | 2.2372↓ | 3.3986↓ |
| CGBA | 1.1233 | 2.1060 | 1.0640 | 0.7503 | 6.6993 | 0.1547 | 1.2142 | 1.4383 | 2.6854 |
| CGBA+SAR | 1.0382↓ | 2.0571↓ | 1.0050↓ | **0.6033↓** | 4.6945↓ | 0.1477↓ | 1.1901↓ | 1.2496↓ | 1.9948↓ |
| TtBA | 0.9969 | 1.5747 | 0.9618 | 0.7351 | 3.8287 | 0.1508 | 1.1936 | 1.2665 | 2.0481 |
| TtBA+SAR | **0.9493↓** | **1.4616↓** | **0.7536↓** | 0.6137↓ | **3.2911↓** | **0.1416↓** | **1.1237↓** | **1.2490↓** | **1.9654↓** |

*Table 8.* Average SSIM across different attack settings.

| Dataset | ImageNet | | | | | CIFAR-10 | | MNIST | |
|---|---|---|---|---|---|---|---|---|---|
| Model | ViT | ResNet | Inception | VGG | Engstrom | CNN | WRN | CNN | LMT |
| HSJA | 0.9758 | 0.8911 | 0.9561 | 0.9838 | 0.8308 | 0.9935 | 0.8554 | 0.4259 | 0.3839 |
| HSJA+SAR | 0.9813↑ | 0.8981↑ | 0.9571↑ | 0.9874↑ | 0.8697↑ | 0.9942↑ | 0.8847↑ | 0.6961↑ | 0.6167↑ |
| CGBA | 0.9937 | 0.9599 | 0.9801 | 0.9964 | 0.9103 | 0.9985 | 0.9637 | 0.5395 | 0.4567 |
| CGBA+SAR | 0.9954↑ | 0.9693↑ | 0.9826↑ | **0.9973↑** | 0.9231↑ | 0.9988↑ | 0.9798↑ | 0.7144↑ | 0.5791↑ |
| TtBA | 0.9949 | 0.9764 | 0.9877 | 0.9966 | 0.9338 | 0.9989 | 0.9691 | 0.7972 | 0.6532 |
| TtBA+SAR | **0.9955↑** | **0.9815↑** | **0.9879↑** | 0.9972↑ | **0.9538↑** | **0.9993↑** | **0.9812↑** | **0.8931↑** | **0.8115↑** |

# D. Importance Maps Generated by SAR

Figure 1 illustrates the importance maps and corresponding perturbations generated by SAR. In the visualization, red regions indicate high sensitivity, blue regions indicate low sensitivity, and yellow regions represent medium sensitivity. We observe that the high-sensitivity areas identified by SAR are consistently concentrated on semantically meaningful objects, such as streetlights, parrots, trucks, and dogs. This suggests that SAR effectively captures the important regions of the image.

| Original images | Importance maps and perturbations generated by SAR | Original images | Importance maps and perturbations generated by SAR |
|---|---|---|---|

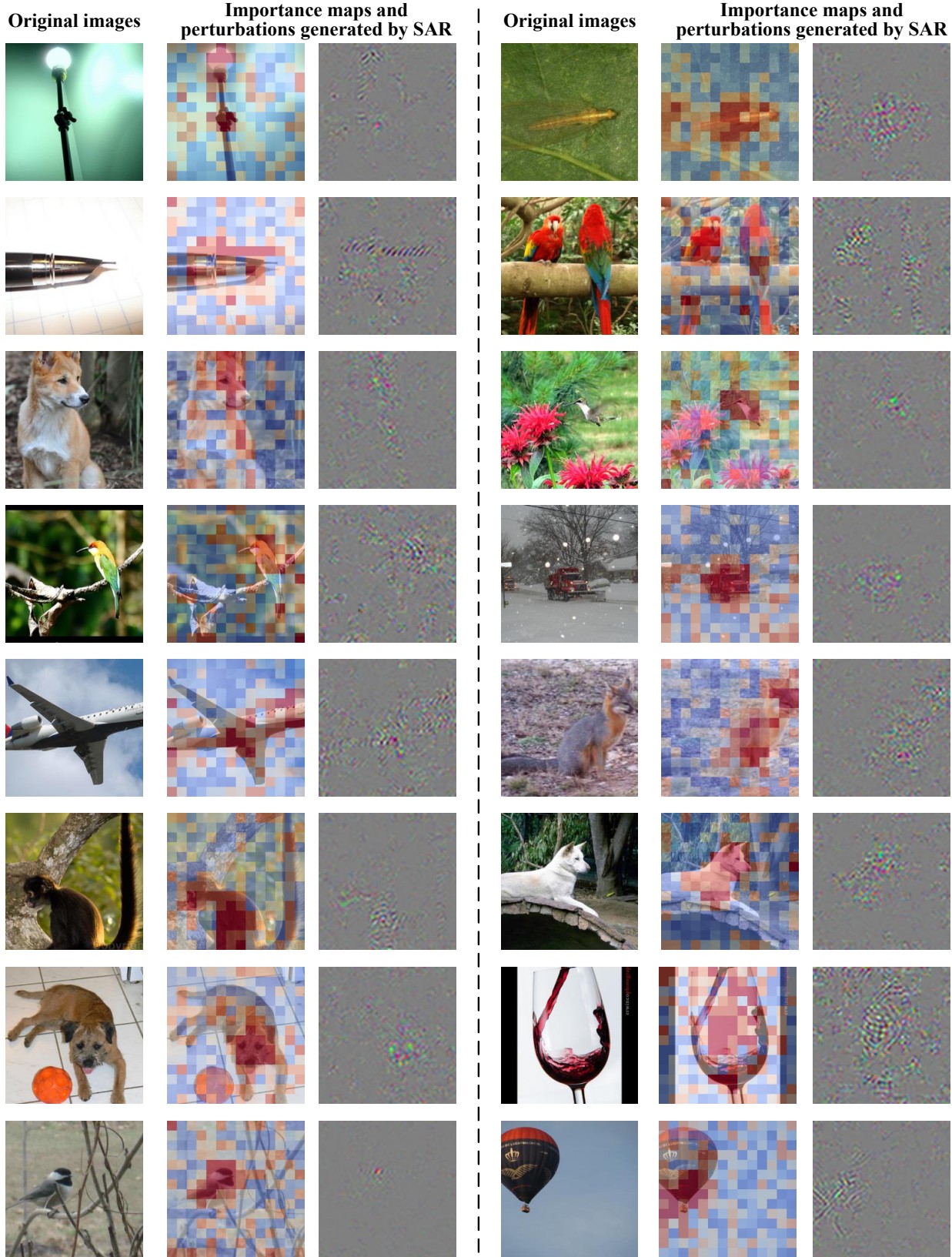

*Figure 1.* Importance maps and corresponding perturbations generated by SAR.

