# OpenReview forum: "Bias in Zeroth-Order Normal Estimation for Decision-Based Attacks"
_ICML.cc/2026/Conference — ICML 2026 regular_

### Official Review · Reviewer_KJ9z · 2026-03-11

**Soundness:** 3
**Presentation:** 3
**Significance:** 2
**Originality:** 3
**Overall Recommendation:** 4
**Confidence:** 4

**Summary:**

The paper analyzes decision-based adversarial attacks that use zeroth-order Monte Carlo probing to estimate decision-boundary normals. It argues that these methods implicitly assume uniform input sensitivity, while in practice only a small subset of coordinates significantly affects model predictions, causing many dimensions to receive noisy updates under limited query budgets. The authors theoretically characterize this effect through a signal-to-noise analysis showing that only O(N) coordinates can be reliably refined with N queries. Motivated by this insight, they propose Sensitivity-Aware Rescaling (SAR), a lightweight refinement method that estimates an importance map from the current perturbation and suppresses low-importance regions to improve attack efficiency.

**Compliance With Llm Reviewing Policy:**

Affirmed.

**Final Justification:**

I thank the authors for the clear and constructive rebuttal. The additional comparison with PAR strengthens the empirical evaluation and addresses one of my main concerns. While some limitations remain regarding the theoretical assumptions and the indirect estimation of sensitivity, the paper is technically sound and demonstrates consistent practical improvements. Overall, I find the contribution valuable and adjust my recommendation to a weak accept.

**Key Questions For Authors:**

Check weaknesses.

**Limitations:**

yes

**Strengths And Weaknesses:**

Strengths:

- The paper provides a stochastic analysis of the Monte Carlo normal estimator, linking its signal-to-noise behavior to heterogeneous input sensitivity, and uses this insight to propose SAR, a simple plug-in refinement method that can be easily integrated into existing decision-based attacks without modifying their core optimization procedures.

Weaknesses:

- The analysis relies on a local linear sign model, isotropic sampling assumptions, and large-dimensional approximations. It is unclear how well these assumptions hold in realistic deep neural network decision boundaries.

-  SAR infers importance using the magnitude of the current perturbation, which may reflect optimization artifacts of the base attack rather than true model sensitivity.

-  While the paper discusses region-sensitive attacks in related work, the experiments do not extensively compare SAR against alternative importance-based or sparse perturbation approaches.

---

> ### Author Rebuttal · Authors · 2026-03-29
>
> Thank you for your important comments.
>
> **Response to Weakness 1:**
> These assumptions are **widely adopted** in existing studies. The local linear sign model is used only as a local approximation under a sufficiently small probing radius [1-2]. Isotropic sampling aligns with the default probing strategy used in Monte Carlo normal estimators [1-2]. The large-dimensional approximation is mainly used to make the analysis more tractable [1-2], without affecting the main qualitative conclusions.
>
> Meanwhile, we agree that the assumptions in our analysis are idealized and may not hold exactly for realistic DNN. Our goal is not to fully characterize real boundary geometry, but to **explain the key local mechanism** underlying Monte Carlo gradient estimation in decision-based attacks. While we acknowledge these limitations, they do not undermine **the value of our theoretical and empirical contributions**. We will further clarify the scope and limitations in the revision.
>
> **Response to Weakness 2:**
> We respectfully clarify that Theorem 4.3 provides the key justification for using the perturbation magnitude in SAR. Specifically, **our theoretical result is rooted in expectation**:  the expected coordinate-wise perturbation magnitude captures **accumulated local sensitivity information**, rather than simply reflecting the artifacts of the base attack’s optimization trajectory. We have also included importance maps generated by SAR in the anonymous code link, which show that the important regions identified by SAR consistently align with semantically meaningful objects, such as streetlights, parrots, trucks, and dogs.
>
> Meanwhile, we agree that in finite-query and non-fully-converged regimes, such artifacts may arise; however, SAR explicitly incorporates smoothing to enhance the robustness of the inferred importance map and mitigate their impact, as demonstrated in the empirical results provided in our response to reviewer pSrH (Response to Weakness 1). Furthermore, the additional experiments in Response to Weakness 3 also show that SAR consistently outperforms the baselines on real-world models.
>
> **Response to Weakness 3:**
> We agree that evaluating SAR against region-sensitive methods is important. Under strict decision-based settings, however, such methods are very limited, as most existing methods rely on gradient or score information. To the best of our knowledge, PAR [3] is one of the few methods that satisfies this constraint while incorporating region-level sensitivity modeling, making it a suitable baseline. PAR performs binary decisions on each region, either keeping or removing its perturbation, representing a coarse, all-or-nothing strategy that contrasts with SAR’s fine-grained approach.
>
> The results below indicate that PAR has limited performance, as it ignores the intrinsic structure of perturbations and relies on coarse, all-or-nothing updates. In contrast, SAR captures fine-grained sensitivity patterns, enabling more precise and effective refinement, resulting in consistently better results.
>
> **Performance comparison on the ImageNet dataset using a ViT model**
> | Attackers | ASR | AVG $\ell_2$ | MID $\ell_2$ | SSIM |
> |-----------|-----|--------------|--------------|------|
> | HSJA        | 62.6% | 2.9681 | 2.2590 | 0.9758 |
> | HSJA+PAR    | 70.0% | 2.7396 | 2.1388 | 0.9793 |
> | HSJA+SAR    | 71.8%| 2.5712 | 1.8941 | 0.9813 |
> | CGBA        | 84.8% | 1.5685 | 1.1233 | 0.9937 |
> | CGBA+PAR    | 89.2%| 1.4584 | 1.1031 | 0.9950 |
> | CGBA+SAR    | 90.4%| **1.3542** | 1.0382 | 0.9954 |
> | TtBA        | 86.0% | 1.5019 | 0.9969 | 0.9949 |
> | TtBA+PAR    | 88.2% | 1.4793 | 0.9967 | 0.9949 |
> | TtBA+SAR    | **90.6%** | 1.3613 | **0.9493** | **0.9955** |
>
> ---
>
> **Performance comparison on the ImageNet dataset using a VGG model**
> | Attackers | ASR | AVG $\ell_2$ | MID $\ell_2$ | SSIM |
> |-----------|-----|--------------|--------------|------|
> | HSJA        | 72.2%| 2.4745 | 1.6251 | 0.9838 |
> | HSJA+PAR    | 79.6%| 2.0897 | 1.3951 | 0.9870 |
> | HSJA+SAR    | 79.6% | 2.0545 | 1.2663 | 0.9874 |
> | CGBA        | 93.0%| 1.1743 | 0.7503 | 0.9964 |
> | CGBA+PAR    | 94.0%| 1.0406 | 0.6475 | 0.9971 |
> | CGBA+SAR    | **95.8%**| **0.9596** | **0.5733** | **0.9973** |
> | TtBA        | 93.0%| 1.1498 | 0.7351 | 0.9966 |
> | TtBA+PAR    | 93.2% | 1.0685 | 0.7096 | 0.9969 |
> | TtBA+SAR    | 94.4%| 0.9829 | 0.6137 | 0.9972 |
>
> [1] Jianbo Chen, Michael I Jordan, and Martin J Wainwright. Hopskipjumpattack: A query-efficient
> decision-based attack. In 2020 IEEE Symposium on Security and Privacy, pp. 1277–1294, 2020a.
>
> [2] Md Farhamdur Reza, Ali Rahmati, Tianfu Wu, and Huaiyu Dai. CGBA: Curvature-aware geometric
> black-box attack. In Proceedings of the IEEE/CVF International Conference on Computer Vision,
> pp. 124–133, 2023.
>
> [3] Yucheng Shi, Yahong Han, Yu-an Tan, and Xiaohui Kuang. Decision-based black-box attack against vision transformers via patch-wise adversarial removal. Advances in Neural Information Processing Systems, 35:12921–12933, 2022.

---

> > ### Author Rebuttal · Reviewer_KJ9z · 2026-04-03
> >
> > I thank the authors for their clear and constructive rebuttal.
> >
> > The additional clarifications helped address several of my concerns. In particular, the comparison with PAR strengthens the empirical evaluation and demonstrates that SAR provides consistent improvements over existing region-sensitive approaches under decision-based constraints.
> >
> > The discussion on the theoretical assumptions and the role of perturbation magnitude as a proxy for sensitivity is reasonable, although some concerns remain regarding the gap between the theoretical analysis and realistic neural network behavior, as well as the indirect nature of the sensitivity estimation.
> >
> > Overall, I find the paper to be technically sound, well-motivated, and practically useful. While some limitations remain, the combination of theoretical insight and consistent empirical improvements makes this a valuable contribution.
> >
> > Accordingly, I adjust my recommendation to a weak accept.

---

> > > ### Author Response · Authors · 2026-04-04
> > >
> > > Thank you very much for the constructive feedback and the positive evaluation of our work. We appreciate your insightful comments which help us clarify the theoretical limitations and better highlight the improvements over the baselines.

---

### Official Review · Reviewer_pSrH · 2026-03-12

**Soundness:** 3
**Presentation:** 4
**Significance:** 3
**Originality:** 4
**Overall Recommendation:** 5
**Confidence:** 4

**Summary:**

The paper shows that zeroth‑order normal estimation in decision‑based attacks has an inherent expectation‑level bias: only a small set of high‑sensitivity coordinates receive meaningful updates, while most dimensions stagnate due to one‑bit feedback and heterogeneous pixel sensitivity. Building on this theoretical characterization, the authors observe that the final perturbation produced by such attacks implicitly encodes a reliable sensitivity ranking, which they exploit through a lightweight post‑processing step called SAR. SAR uses that sensitivity signal to systematically shrink low‑impact regions and reproject onto the boundary, producing lower‑norm and more visually coherent adversarial examples without modifying the underlying attack pipeline.

**Compliance With Llm Reviewing Policy:**

Affirmed.

**Final Justification:**

Main concerns with respect to remaining in valid region for reprojection have been addressed by additional experiments, and concerns for overly aggressive SAR have been mitigated by clarifications. Raising score.

**Key Questions For Authors:**

Since SAR modifies the perturbation independently of the probing radius used during normal estimation, could large suppressions violate the local linearity assumption required for the boundary projection step? If so, how do you detect or mitigate such cases?

**Limitations:**

Yes

**Strengths And Weaknesses:**

Strengths

The paper presents a mathematically precise characterization of how heterogeneous pixel sensitivities induce an expectation‑level bias in Monte‑Carlo normal estimation, giving the method a solid theoretical backbone.

The authors' normal‑estimation routine is simple, fully vectorizable, and naturally fits modern batch‑compute settings, making the overall attack easy to parallelize without complicating the pipeline.

SAR is a low‑overhead refinement step that leverages the stationary sensitivity pattern without altering the attack pipeline, making it easy to combine with existing attack methods.

Weaknesses

SAR can only operate after an existing decision‑based attack has already produced a reasonably accurate boundary point, so its benefits vanish when the base attack does not converge well.

Because SAR’s suppression step is independent of the normal estimator’s probing radius or local linearity region, an overly aggressive reduction of low‑sensitivity pixels risks moving the perturbation outside the region where the earlier boundary normal remains geometrically valid for reprojection.

---

> ### Author Rebuttal · Authors · 2026-03-29
>
> Thank you for your inspiring and valuable comments.
>
> **Response to Weakness 1:**
>
> We note that SAR builds on a base decision-based attack that has produced a reasonably accurate boundary point, such that the resulting perturbation encodes meaningful sensitivity information. When the base attack is less well converged (e.g., under very low query budgets with unstable perturbations), the inferred importance map becomes less reliable, potentially resulting in reduced effectiveness.
>
> However, our additional experiments provide strong evidence that **SAR remains highly effective** in the practically relevant regime **where the base attack is not yet fully converged** but has already captured meaningful partial sensitivity information. Specifically, even with a query budget of 5,000, where the base attack is noticeably less converged than under 10,000 queries, SAR can manage to deliver substantial and consistent reductions in the $\ell_2$ norm.
>
> **Average $\ell_2$ performance under 5,000 query budget**
> | Dataset | ImageNet | ImageNet | CIFAR-10 | MNIST |
> |---------|----------|----------|----------|--------|
> | Model   | ViT      | VGG      | CNN      | CNN    |
> | HSJA        | 5.3775 | 4.2073 | 0.7381 | 3.6559 |
> | HSJA+SAR    | 5.2384↓ | 4.1698↓ | 0.7205↓ | 3.4293↓ |
> | CGBA        | 2.4082 | 1.6904 | 0.2140 | 1.7049 |
> | CGBA+SAR    | **2.0098↓** | **1.4994↓** | 0.1998↓ | 1.5793↓ |
> | TtBA        | 2.1927 | 1.7648 | 0.2250 | 2.2721 |
> | TtBA+SAR    | 2.0699↓ | 1.5610↓ | **0.1980↓** | **1.3535↓** |
>
> This robustness primarily arises from the average pooling and upsampling steps in SAR, which make the inferred importance map more resilient to noise. We also note that only under extremely small query budgets (e.g., around 2,000 queries), where the base attack has not yet accumulated sufficient sensitivity information, the effectiveness of SAR may be reduced.
>
> **Response to Weakness 2 and Question 1**:
> We agree that overly aggressive suppression may push the perturbation outside the local region where the boundary geometry remains valid, causing reprojection to land on a distant boundary.
>
> However, **SAR does not rely on the local linearity assumption used in normal estimation**. Instead, its effectiveness stems from a different principle: after reprojection, the perturbation magnitude continues to preserve the sensitivity ranking across coordinates. This is the central insight formalized in Theorem 4.3. We will explicitly clarify this point in the revised paper.
>
> In practice, this assumption typically holds, as SAR is applied once the base attack has entered a stable regime. At this stage, the adversarial example is already close to the original image, the perturbation remains small, and the boundary geometry is sufficiently well-behaved for reprojection, making the resulting sensitivity ranking reliable in real-world settings.
>
> Moreover, SAR includes several safeguards against overly large suppression.
> First, each update is conservative, using a fixed suppression factor $s=0.1$, resulting in gradual, cumulative redistribution rather than a single aggressive reduction.
> Second, after each suppression step, SAR verifies whether the candidate remains adversarial (Algorithm 1, line 15), i.e., $I(x + d_{temp}) = 1$, before performing boundary projection. If the suppression is overly aggressive, the candidate will be discarded.
>
> Finally, we respectfully clarify that we do not directly tie suppression to the probing radius $\delta$ because $\delta$ is chosen for accurate local probing and is extremely small in practice (e.g., $3\times10^{-4}$); using it as the suppression step would make each SAR update nearly negligible and therefore query-inefficient. Prior work such as HSJA [1] adopts a similarly small $\delta$ for local probing, but likewise does not use it as the optimization step size. Instead, HSJA employs a heuristic initial step size $\|perturbation\|_2 \cdot \sqrt{t}$ (where $t$ is the iteration number), followed by a binary search to determine an appropriate step size.
>
> We sincerely thank the reviewer again for this valuable discussion and will revise the paper to clarify these points.
>
> [1] Chen J., Jordan M. I., and Wainwright M. J. Hopskipjumpattack: A query-efficient decision-based attack. In 2020 IEEE Symposium on Security and Privacy, pp.1277–1294, 2020a.

---

> > ### Author Rebuttal · Reviewer_pSrH · 2026-04-05
> >
> > The additional experiments and clarifications with respect to overly large suppression safeguards have addressed my concerns. I will raise my score.

---

> > > ### Author Response · Authors · 2026-04-06
> > >
> > > Thank you very much. We sincerely thank the reviewer again for this valuable discussion, and we greatly appreciate your positive reassessment of our work.

---

### Official Review · Reviewer_JG9E · 2026-03-13

**Soundness:** 3
**Presentation:** 2
**Significance:** 3
**Originality:** 3
**Overall Recommendation:** 5
**Confidence:** 3

**Summary:**

The paper presents theoretical results regarding the distribution of Monte Carlo estimates of the decision boundary normal vector used in zeroth-order optimization for black box adversarial attacks. The results indicate that the estimates of components with low gradient magnitude are dominated by the variance, and hence have a low signal-to-noise ratio. The authors present an algorithm, Sensitivity-Aware Rescaling (SAR), which reduces the $l_2$ norm of the adversarial perturbation by reducing weight on these components.

**Compliance With Llm Reviewing Policy:**

Affirmed.

**Final Justification:**

SAR improves attack success rates and perturbation norms for decision-based attacks, and its core rescaling technique can easily be combined with existing approaches. The experimental results in the rebuttal clarified the motivation for key design choices in SAR. It will be valuable for future work on black-box adversarial attacks.

**Key Questions For Authors:**

Why were the key design choices in SAR made?

Are these theoretically motivated or by empirical performance?

What is the performance of SAR without them?

**Limitations:**

Yes

**Strengths And Weaknesses:**

## Strengths

- The empirical results show clear evidence that SAR improves the performance of decision-based attacks, including state-of-the-art attacks, on realistic datasets and models.
- The rescaling approach is motivated by the theoretical results.

## Weaknesses

- Certain choices made by SAR are not fully examined; in particular, why is the importance map per-pixel, discarding channel information? Why was it necessary to smooth the importance map via average pooling + upsampling? Is this related to uncertainty in the sensitivity estimates?
    - An ablation of these choices would improve the evaluation.
- The writing can be improved. In the abstract, line 29, "this stationarity" has not yet been mentioned. The introduction overuses acronyms for techniques, they should be introduced with their full names, similarly for related work. "strictly non-zero" is used where "magnitude above a threshold" appears to be meant on line 235, and duplicate "a".

## Minor Comments

Equation (7) is not true by definition, and depends on whether the adversarial region is a closed set.

---

> ### Author Rebuttal · Authors · 2026-03-29
>
> Thank you very much for your valuable comments.
>
> **Reply to Weakness 1:**
> We agree that the design choices in SAR require thorough examination.
>
> (1) For the importance map design, we model it at the pixel level rather than incorporating channel-wise information. This choice is motivated by both prior work and empirical evidence. Existing methods such as PAR [1], SaliencyAttack [2], and SRA [3] consistently show that the sensitivities across the three channels are highly correlated, with channels tending to exhibit uniformly high or low sensitivity simultaneously. Therefore, modeling channel-wise importance may offer only marginal additional benefit while incurring higher query cost. Following your helpful suggestion, our additional experiments (shown below, where the first column reports the performance of the channel-wise importance map) further support this design choice.
> At the same polling factor $r=1$, pixel-wise importance map achieves better average (median) $\ell_2$ performance than the channel-wise variant.
>
> | Polling factor $r$ | 1 (3 channels) | 1 | 2 | 4 | 8 | 16 | 32 |
> |-------------------|---------------|---|---|---|---|----|----|
> | VGG+TtBA          | 1.43(1.07)    | 1.39(1.05) | 1.38(1.03) | 1.36(1.00) | 1.34(0.96) | **1.29(0.94)** | 1.29(0.95) |
> | ViT+TtBA          | 1.67(1.13)    | 1.61(1.12) | 1.60(1.11) | 1.56(1.10) | 1.52(1.10) | **1.51(1.08)** | 1.58(1.09) |
> | VGG+HSJA          | 3.83(2.63)    | 3.82(2.62) | 3.62(2.28) | 3.53(2.19) | 3.47(2.19) | **3.41(2.18)** | 3.43(2.24) |
>
> (2) Regarding the necessity of **smoothing the importance map**, this design is motivated by the inherent uncertainty in sensitivity estimation in decision-based attacks, where Monte Carlo probing introduces significant noise. Applying average pooling followed by upsampling helps reduce this noise and yields a more stable and reliable importance map for guiding the attack.
>
> We have discussed this point in Section 5 (line 283), but we agree it should be made clearer as you pointed out. In addition, Appendix B (line 762) provides parameter sensitivity and ablation studies. As shown in Table 3 (b–d) and the Table above, using an appropriate pooling factor $r$ consistently improves SAR’s performance compared to the case without smoothing (i.e., $r$=1).
>
> [1] Yucheng Shi, Yahong Han, Yu-an Tan, and Xiaohui Kuang. Decision-based black-box attack against vision transformers via patch-wise adversarial removal. Advances in Neural Information Processing Systems, 35:12921–12933, 2022.
>
> [2] Zeyu Dai, Shengcai Liu, Qing Li, and Ke Tang. Saliency attack: Towards imperceptible black-box adversarial attack. ACM Transactions on Intelligent Systems and Technology, 14(3):1–20, 2023.
>
> [3] Chenhao Lin, Sicong Han, Jiongli Zhu, Qian Li, Chao Shen, Youwei Zhang, and Xiaohong Guan. Sensitive region-aware black-box adversarial attacks. Information Sciences, 637:118929, 2023.
>
> **Reply to Weakness 2**: Thank you again for your helpful comments on the writing. We have revised the Abstract (line 26) to more clearly introduce the notion of stationarity. Specifically, we now state: "By modeling ZO refinement as a stochastic dynamical system, we characterize its long-term behavior: the optimization enters a **stationary regime** ..."
>
> We also revised the introduction and related work sections to introduce all acronyms with their full names at first occurrence.
> Finally, we corrected the wording and the duplicated article typo on line 235 as "only $O(N)$ input coordinates can exhibit a non-negligible expected drift".
>
> **Reply to Minor Comments**: We agree that Equation (7) does not follow directly from the definition and instead depends on whether the adversarial region is closed along the search ray. We have revised the paper accordingly: "Under the assumption that the adversarial region is closed along the search ray, $\tilde{x}^*(d)$ lies on the boundary, satisfying ... "
>
> **Reply to Question 1, 2, and 3**: The key design choices in SAR are motivated by both theoretical considerations and empirical performance.
>
> (1) For the design choice of defining the importance map **per-pixel** rather than channel-wise, our motivation is supported by both prior findings in related work and our empirical results (see Reply to Weakness 1).
>
> (2) For the design choice of smoothing the importance map via **average pooling + upsampling**, this is motivated both theoretically and empirically. Proposition 4.2 indicates that sensitivity estimates are inherently noisy, making importance maps unstable. Average pooling with upsampling mitigates local fluctuations and enforces spatial consistency. Our ablation results (Reply to Weakness 1) also confirm that this smoothing consistently improves SAR’s performance.
>
> (3) Even without these design choices, SAR remains functional, but its performance degrades noticeably. This degradation is consistently observed in the ablation studies, including the newly added results in Reply to Weakness 1 and Appendix B (line 762).

---

> > ### Author Rebuttal · Reviewer_JG9E · 2026-04-04
> >
> > Thank you for the additional experiments and clarifications, my concerns are resolved; I will raise my score.

---

> > > ### Author Response · Authors · 2026-04-04
> > >
> > > We sincerely thank the reviewer for carefully evaluating our revisions and for the encouraging feedback. Your suggestions help us a lot to clarify the motivation of our method.

---

### Decision · Program_Chairs · 2026-04-30

**Decision:**

Accept (regular)

**Comment:**

All reviewers acknowledge that the rebuttal has addressed most of the concerns and recommend acceptance. The AC has read all the reviews and the authors' responses, and the recommendation is acceptance.